# One Year of Recombinant Human Growth Hormone Treatment in Adults with Prader–Willi Syndrome Improves Body Composition, Motor Skills and Brain Functional Activity in the Cerebellum

**DOI:** 10.3390/jcm11071831

**Published:** 2022-03-25

**Authors:** Laia Casamitjana, Laura Blanco-Hinojo, Olga Giménez-Palop, Jesús Pujol, Gerard Martínez-Vilavella, Susanna Esteba-Castillo, Rocío Pareja, Valentín Freijo, Laura Vigil, Joan Deus, Assumpta Caixàs

**Affiliations:** 1Endocrinology and Nutrition Department, Hospital Universitari Parc Taulí, 08208 Sabadell, Spain; lcasamitjana@tauli.cat (L.C.); ogimenez@tauli.cat (O.G.-P.); rpareja@tauli.cat (R.P.); 2Department of Medicine, Universitat Autònoma de Barcelona, 08208 Sabadell, Spain; vfreijo@tauli.cat; 3Institut d’Investigació i Innovació Parc Taulí (I3PT)-CERCA, 08208 Sabadell, Spain; lvigil@tauli.cat; 4Centro Investigación Biomédica en Red de Salud Mental, CIBERSAM G21, 08003 Barcelona, Spain; laura.blanco02@gmail.com (L.B.-H.); 21404jpn@comb.cat (J.P.); 5MRI Research Unit, Department of Radiology, Hospital del Mar, 08003 Barcelona, Spain; g.martinezvilavella@gmail.com (G.M.-V.); joan.deus@uab.cat (J.D.); 6Specialized Service in Mental Health and Intellectual Disability, Institut Assistència Sanitària (IAS), Parc Hospitalari Martí i Julià, 17190 Girona, Spain; susanna.esteba@ias.cat; 7Neurodevelopment Group [Girona Biomedical Research Institute (IDIBGI)-IAS], 17190 Girona, Spain; 8Physical Medicine and Rehabilitation Department, Parc Taulí Hospital Universitari, 08208 Sabadell, Spain; 9Pneumology Department, Hospital Universitari Parc Taulí, 08208 Sabadell, Spain; 10Department of Clinical and Health Psychology, Autonomous University of Barcelona, 08193 Bellaterra, Spain

**Keywords:** Prader–Willi syndrome, growth hormone, hypotonia, motor function, fMRI, myokines

## Abstract

We compared body composition, biochemical parameters, motor function, and brain neural activation in 27 adults with Prader–Willi syndrome and growth-hormone deficiency versus age-and sex-matched controls and baseline versus posttreatment values of these parameters after one year of recombinant human growth hormone (rhGH) treatment. To study body composition, we analyzed percentage of fat mass, percentage of lean mass, and muscle-mass surrogate variables from dual X-ray absorptiometry. Biochemical parameters analyzed included IGF-I, glucose metabolism, and myokines (myostatin, irisin, and IL6). To explore muscle function, we used dynamometer-measured handgrip strength, the Timed Up and Go (TUG) test, and the Berg Balance Scale (BBS). To study brain activation, we acquired functional magnetic resonance images during three motor tasks of varying complexity. After one year of treatment, we observed an increase in lean mass and its surrogates, a decrease in fat mass, improvements in TUG test and BBS scores, and increased neural activation in certain cerebellar areas. The treatment did not significantly worsen glucose metabolism, and no side-effects were reported. Our findings support the benefits of rhGH treatment in adults with Prader–Willi syndrome and growth-hormone deficiency on body composition and suggest that it may also improve balance and brain neural activation.

## 1. Introduction

Prader–Willi syndrome (PWS) is a rare chromosomal disease affecting multiple systems, the most characteristic manifestation being hypotonia from birth. In early childhood, subjects develop hyperphagia that can lead to morbid obesity [1]. Other characteristics of PWS include dysmorphic alterations, behavioral disturbances, and cognitive impairment [2,3], as well as endocrine dysfunctions, such as growth hormone (GH) deficiency (GHD) leading to short stature, hypogonadism, and hypothyroidism [4].

Motor performance is especially affected in childhood. Although severe hypotonia improves several weeks after birth, a degree of hypotonia and muscle weakness remain, resulting in severely delayed motor development [1,5]. Motor problems persist throughout childhood into adulthood, and adults with PWS score well below the normal range on standardized motor performance tests [6,7,8,9,10].

The causes of hypotonia, muscle weakness, and motor problems in patients with PWS are not clear. It is presumed that one contributing factor would be abnormal body composition (increased fat mass and decreased muscle mass), thought to be related to hormonal deficiencies arising from hypothalamic dysfunction. The thyroidal, gonadal, adrenal-cortex, and GH axes can be affected; the GH axis has the greatest impact on body composition [11,12,13]. Another possible contributing factor might be neuromuscular anomalies [6]; qualitative biopsy studies in infants with PWS have revealed some abnormalities in muscle tissue [5,14]. On the other hand, the functioning of the motor cortex seems to be impaired in adult patients with PWS, who show hypo-excitability of these areas [15] and weaker activation in the cerebellum during motor coordination compared with controls [16]. Moreover, widespread structural and volumetric abnormalities in the brain may also affect some specific motor aspects [17].

Recombinant human growth hormone (rhGH) treatment in adults with PWS induces metabolically beneficial changes in body composition, increasing lean mass and decreasing fat mass [18,19,20]. Various authors have explored the amelioration effects of rhGH treatment on motor performance, hypothesizing that increases in lean mass (mainly composed of muscle mass) would result in improved motor performance and finding increased strength [18,21,22] and exercise capacity [18,23,24], but no improvement in balance [18,22]. These effects were independent of GHD prior to treatment [18,22,23]. However, studies on this issue are scarce [25], and no studies have specifically evaluated functional abnormalities in motor and coordination brain areas in adults with PWS before and after rhGH treatment.

Myokines are cytokines produced and released by skeletal muscle cells in response to muscular contractions [26]. Different myokines have different effects on muscle mass. Myostatin negatively regulates muscle growth [27]. By contrast, irisin not only positively regulates muscle growth, but also can convert white adipose tissue to brown fat [28], increasing energy expenditure and improving glucose tolerance. Discordant myokine levels have been reported in studies in patients with PWS [29,30,31,32].

We sought, first, to characterize body composition, strength, balance, myokines, and brain neural activation during the performance of different motor tasks in adults with PWS and GHD by comparing these parameters with healthy controls and, second, to determine the effects of one year of rhGH treatment on these parameters by comparing baseline and posttreatment values. We hypothesized that treatment would improve body composition (increasing lean mass and decreasing fat mass), increase balance and strength, and improve brain neural activation during motor tasks.

## 2. Materials and Methods

### 2.1. Participants

We selected patients treated at our center between 1 January 2016, and 31 January 2019 aged ≥ 18 years with genetically diagnosed PWS and GHD diagnosed with the growth-hormone-releasing hormone (GHRH)-arginine test and/or the glucagon test, using the following cutoffs: GH < 11 ng/mL if body-mass index (BMI) < 25 kg/m^2^, <8 ng/mL if BMI 25–30 kg/m^2^, and <4 ng/mL if BMI ≥ 30 kg/m^2^ on the GHRH-arginine test and GH < 3 ng/mL at any timepoint on the glucagon test [33]. We excluded patients with uncontrolled diabetes, untreated severe obstructive sleep apnea, active malignancy, or active psychosis. To compose the control group, we matched healthy lean subjects by age and sex. 

The study complied with all provisions in the Declaration of Helsinki and was approved by the local ethics committee. All patients with PWS agreed to participate after being informed together with their parents or caregivers; their legal guardians (usually their parents) signed the consent form before enrollment. All control participants provided written informed consent.

### 2.2. Study Design

We compared baseline variables between groups (patients with PWS vs. controls). Subjects with PWS started recombinant rhGH therapy (Genotonorm Miniquick^®^, Pfizer, New York, NY, USA) at an initial dose of 0.2 mg/day. To avoid overdosing, the dose was periodically adjusted to maintain serum total IGF-I within 2 standard deviations of an age-matched reference value. The maximum dose was 0.6 mg/day in men and 0.8 mg/day in women. After 12 months’ rhGH therapy, measures were repeated in the PWS group and compared with baseline values.

#### 2.2.1. Anthropometric Methods

Physical examination included measurements of height (Stadiometer Harpenden, Holtain Ltd., Dyfed, UK), weight (measured to the nearest 0.1 kg with standard equipment), and waist circumference measured halfway between the costal edge and iliac crest. BMI was calculated as weight in kilograms divided by the height in meters squared. According to the World Health Organization criteria (https://www.who.int/news-room/fact-sheets/detail/obesity-and-overweight, accessed on 9 February 2022), BMI was classified as normal (18.5–24.9 kg/m^2^), overweight (25–29.9 kg/m^2^), or obese (≥30 kg/m^2^).

#### 2.2.2. Body Composition Assessment

Body composition was assessed by dual energy X-ray absorptiometry (DXA) (Lunar Prodigy-963, Chicago, IL, USA). The variables considered in the DXA analysis were percentage of lean mass, percentage of fat mass, and appendicular lean mass (ALM, Kg) as a surrogate variable for skeletal muscle mass. ALM was adjusted for weight (ALM/weight) and BMI (ALM/BMI) by dividing it by these two variables.

#### 2.2.3. Metabolic Evaluation

IGF-I, glucose, glycosylated hemoglobin (HbA1c), irisin, myostatin, and IL-6 were determined in fasting blood samples.

#### 2.2.4. Motor Function Assessment

Motor performance was evaluated with three tests. As an indicator of general muscle strength, we measured handgrip strength with a hydraulic hand dynamometer (Jamar, Saehan Corporation, Changwon, Korea). Subjects sat in a chair with their feet touching the ground, shoulder adducted and neutrally rotated, with the elbow bent to 90°, the arm against the trunk, and wrist in a neutral position [34]. Grip strength of each hand was measured three times in each of five handle positions. Participants applied the maximum force they could for 3 s, resting 15 s between measurements. We recorded the highest value of grip force measured in each handle position among the 6 attempts and the maximum grip strength, defined as the highest value obtained among the 30 measurements. Functional mobility and fall risk were assessed using the Timed Up and Go (TUG) test [35]. Subjects were asked to sit upright with their back against an armless chair, stand up when instructed, walk three meters at a self-selected pace, turn around, walk back to the chair, and sit back down to the starting position. We determined the total time (in seconds) required to complete the walk, measured with a manual stopwatch, and we recorded the mean value obtained in three trials. To determine subjects’ ability to safely balance during a series of predetermined tasks, we used the Berg Balance Scale (BBS), a list of 14 items rated on a five-point ordinal scale (0–4; 0 = lowest level of function; maximum total score = 56) that takes approximately 20 min to complete; a total score < 40 is commonly used to identify patients at high risk for falls [36].

#### 2.2.5. Functional Magnetic Resonance Imaging (fMRI) Testing

A 3.0 T MRI scanner (Achieva, Philips Healthcare, Best, The Netherlands) equipped with an eight-channel phased-array head coil and single-shot echoplanar imaging (EPI) software was used for fMRI to examine brain neural activation during the performance of three manual tasks of different motor complexity. Details of the examination procedure, fMRI acquisition, processing, and analyses, and the control of potential head motion effects are described elsewhere [16] and in the Appendix A. Motor tasks during fMRI involved (i) repetitive self-paced opening and closing of the hand (unimanual flexion-extension test), alternating hands for 30 s each; (ii) repetitive flexion and extension of the two hands with a phase shift of 180 between them (bimanual anti-phase flexion-extension test); and (iii) a repetitive sequence of fingers-to-thumb opposition movements with the right hand (thumb-opposition test). The fMRI study acquired data in identical blocks in which three 30 s baseline periods alternated with three 30 s periods of movement. Each functional time series consisted of 90 consecutive image sets or volumes obtained over 3 min. All image sequences were inspected for potential acquisition and normalization artifacts.

The baseline data of patients with PWS included in this study were reported in an earlier study [16]. Valid fMRI assessments at both baseline and 12 months after the initiation of rhGH treatment were available for 22 of the patients with PWS.

From the initial 22-patient sample available after 12 months of rhGH treatment, data from one patient were excluded from the analysis of the simple task and data from another patient were excluded in the analysis of the bimanual task owing to excessive head movement during fMRI acquisition.

#### 2.2.6. Polysomnography

All patients underwent a complete polysomnography before and after GH treatment. At baseline, 8 patients were under treatment with positive pressure devices (6 with continuous positive airway pressure and 2 with non-invasive ventilation). Polysomnography (Alice 6, Philips Respironics, Murrysville, PA, USA) was performed in a sleep laboratory using a standardized procedure. Sleep was staged in 30-second epochs using the American Academy of Sleep Medicine criteria [37]. Apnea events were scored if nasal pressure or flow decreased by ≥90% from baseline for ≥10 s. Hypopneas were scored when the peak signal excursions dropped by ≥30% of pre-event baseline using nasal pressure during ≥10 s or was associated with an arousal. Events were scored as central if apnea was associated with absent inspiratory effort throughout the entire period of absent airflow. Otherwise, events were scored as obstructive. Indices were calculated by dividing the number of events by the hours of sleep to standardize event numbers for different total sleep times.

### 2.3. Statistical Analyses

To determine whether variables were normally distributed, we used the Shapiro-Wilk test. For comparisons between the PWS and control groups, we used Student’s *t*-test for normally distributed continuous variables, the Mann–Whitney test for non-normally distributed continuous variables, and Pearson’s chi-squared test for categorical variables.

For comparisons between baseline and post-rhGH-treatment assessments in subjects with PWS, we used Student’s *t*-test for normally distributed continuous variables and the Wilcoxon signed-rank test for non-normally distributed continuous variables. We used Pearson’s and Spearman’s correlation coefficients to explore associations among variables. *p* values < 0.05 were considered significant. All analyses were carried out with SAS v9.4 software (SAS Institute Inc., Cary, NC, USA).

Imaging data were processed using MATLAB version 2016a (The MathWorks Inc, Natick, MA, USA) and Statistical Parametric Mapping (SPM12; The Wellcome Department of Imaging Neuroscience, London). In the first-level (single-subject) analysis, we modeled brain activation with a boxcar regressor considering three activation blocks of 30 s and applying a hemodynamic delay of 4 s. Brain activation was estimated based on the contrasts “baseline blocks < motion blocks”. The resulting first-level SPM contrast images were carried forward to subsequent group-level analyses. Single-sample *t*-test designs were used to generate group activation maps from the data acquired 12 months after initiating rh-GH treatment, and paired *t*-tests were used to compare brain activity between baseline and post-rhGH treatment in each of the three motor tests, including data from 20 participants in the first and second task analyses and 22 participants in the third task analysis. In addition, SPM whole-brain regression models were used to map, voxel-wise, the correlation between changes in brain activation between baseline and 12 months posttreatment and changes in individual ratings in the motor tests, analytical, and body composition parameters that had changed significantly after 12 months (i.e., TUG and BBS tests, IGF-I, and percentage of fat and lean mass). In all analyses, results were considered significant with clusters > 2.3 mL at a height threshold of *p* < 0.005, which satisfied the family-wise error (FWE) rate correction of *p*_FWE_ < 0.05.

## 3. Results

We included 27 patients with PWS (15 women; median age, 26 y (IQR 24.0–37.0)). Of these, 14 (52%) had undergone rhGH therapy in childhood. Cytogenetic analysis revealed that 17 had microdeletions (7 type I, 10 type II), 6 had maternal uniparental disomy, 3 had imprinting defects, and 1 had an atypical deletion from locus MICRN3 (BP2) to APAB2 (BP4). In total, 6 patients had type 2 diabetes with a good glycemic control (HbA1c < 7.5%); of these, one was treated only with metformin, two with metformin plus DPPIV inhibitor plus insulin secretagogue, and three with metformin plus DPPIV inhibitor plus insulin. A total of 15 (7 women) were receiving sex steroids.

The control group comprised 22 volunteers (13 women; median age, 27.5 y (IQR 22.0–27.0)) recruited from hospital staff or acquaintances.

### 3.1. Comparison of Baseline Measurements in the PWS and Control Groups

At baseline, the two groups did not differ in sex or age. Weight, BMI, and waist circumference were higher in the PWS group. Body composition analysis showed a higher percentage of fat mass and a lower percentage of lean mass in the PWS group; ALM/weight and ALM/BMI were also lower in the PWS group. Compared with the control group, the PWS group had higher plasma glucose, HbA1c, and irisin and lower insulin-like growth factor (IGF-I). Myostatin and IL-6 did not differ between groups. Handgrip strength was lower in the PWS group. The ability to balance safely measured by the BBS was worse in the PWS group, although only one patient was considered at risk for falling (BBS < 40). The PWS group required more time than the control group to perform the tasks in the TUG test (Table 1).

### 3.2. Comparison between Parameters Measured at Baseline and 12 Months after the Initiation of rhGH Treatment in the PWS Group

After 12 months’ rhGH treatment, no significant changes were observed in weight, BMI, or waist circumference. The percentage of fat mass was significantly lower (median difference, −1.7%; 95%CI: −2.3, −0.2), and the percentage of lean mass was significantly higher (median difference, 2%; 95%CI: 0.50, 2.90). Unadjusted appendicular lean mass remained unchanged, but significant increases were observed in weight-adjusted ALM (median difference, 0.00; 95%CI: 0.00, 0.02) and BMI-adjusted ALM (median difference, 0.01; 95%CI: 0.00, 0.04). IGF-I levels were significantly higher. Glucose and HbA1c were unchanged; in all six patients with type 2 diabetes metabolic control worsened slightly, requiring treatment adjustments (a second oral agent was added in 1, insulin treatment was started in 2, and insulin dose was increased by 0.1 U/kg/day in 3) to maintain strict control. Plasma irisin, myostatin, and IL-6 did not change significantly. No significant worsening of apnea was observed (Table 2). Handgrip strength remained unchanged after treatment. After treatment, less time was required to perform the tasks in the TUG test (median difference, −0.54 s; 95%CI: −1.28, 0.20), and BBS scores were higher (median increase, 1.0 point; 95%CI: 0.00, 2.00) (Table 3). Paired comparisons of imaging data found a significant difference in brain activation during bimanual anti-phase flexion-extension movements (i.e., differences between activation during the task and at rest). Both at baseline and after treatment, fMRI showed activation in the sensorimotor, premotor, and supplementary motor cortices; central operculum; subcortical areas; and cerebellum. After treatment, neural activation in the cerebellar cortex increased significantly (peak difference at MNI coordinates x = −14, y = −46, z = −20; *t* = 4.6, *p* = 0.0001) in lobules IV/V and VI surrounding the primary fissure. Figure 1 illustrates the motor-related activation seen after treatment and within-group differences (baseline vs. 12 m after initiating treatment).

**Table 2 jcm-11-01831-t002:** Anthropometric, biochemical, body composition and polysomnography parameters at baseline and 12 months after the initiation of rhGH therapy in subjects with PWS.

	Baseline (*n* = 27) Median (IQR)	After 12 Months rhGH Treatment (*n* = 27) Median (IQR)	*p*-Value *
Weight (kg)	89.6 (70.5–105.5)	87.7 (74.3–100.2)	0.8091
BMI (kg/m^2^)	34.5 (31.1–41.3)	34.0 (31.8–41.6)	0.8615
WC (cm)	110 (101–124)	112 (104–123.5)	0.6103
IGF-I (ng/mL)	143 (95–188)	217 (160–254)	<0.0001
Glucose (mmol/L)	5.00 (4.50–6.99)	4.83 (4.33–5.49)	0.1085
HbA1c (%)	5.60 (5.30–6.90)	5.60 (5.30–6.20)	0.2384
Irisin (ng/mL)	982.3 (519.4–1789.6)	906.8 (583.5–1770.4)	0.7614
Myostatin (ng/mL)	4.8 (2.95–7.72)	5.57 (3.35–7.37)	0.2176
IL-6 (pg/mL)	15.4(3.58–66.4)	19.9 (3.44–75.0)	0.1948
Fat mass (%)	56.3 (49.3–61.1)	52.1 (49.9–59.2)	0.0053
Lean mass (%)	43.7 (38.9–50.1)	47.9 (40.8–50.1)	0.0009
ALM/weight	0.19 (0.16–0.21)	0.19 (0.17–0.21)	0.0439
ALM/BMI	0.48 (0.38–0.55)	0.49 (0.38–0.56)	0.0262
AHI (*n*/hour)	15.2 (9.20–22.4)	22.3 (7.45–29.4)	0.0911
Central apnea (*n*/TST)	1.0 (0.0–2.0)	0.0 (0.0–2.0)	0.210
Obstructive apnea (*n*/TST)	7.5 (2.5–11.5)	2.0 (1.0–10.25)	0.450
Hypoapnea (*n*/TST)	75.0 (46.0–136.0)	81.0 (28.75–150.25)	0.100

PWS: Prader–Willi Syndrome; IQR: interquartile range; BMI: body mass index; WC: waist circumference; IGF-I: insulin-like growth factor-1; HbA1c: glycosylated hemoglobin; ALM: appendicular lean mass; AHI: apnea–hypopnea Index; *n*/TST: number/total sleep time * Student’s *t*-test for normally distributed variables and Wilcoxon signed-rank test for non-normally distributed variables.

**Table 3 jcm-11-01831-t003:** Motor function at baseline and 12 months after the initiation of rhGH therapy in subjects with PWS.

	Baseline (*n* = 27) Median (IQR)	After 12 Months rhGH Treatment (*n* = 27) Median (IQR)	*p*-Value *
Handgrip strength (kg)	18 (13–22)	16 (13–18)	0.0750
TUG (seconds)	8.76 (7.56–11.9)	8.40 (7.11–9.75)	0.0167
BBS (points)	53 (49–54)	55 (50–55)	0.0208

PWS: Prader–Willi Syndrome; IQR: interquartile range; BBS: Berg Balance Scale; TUG: Timed Up and Go test. * Student’s *t*-test for normally distributed variables and Wilcoxon signed-rank test for non-normally distributed variables.

**Figure 1 jcm-11-01831-f001:**
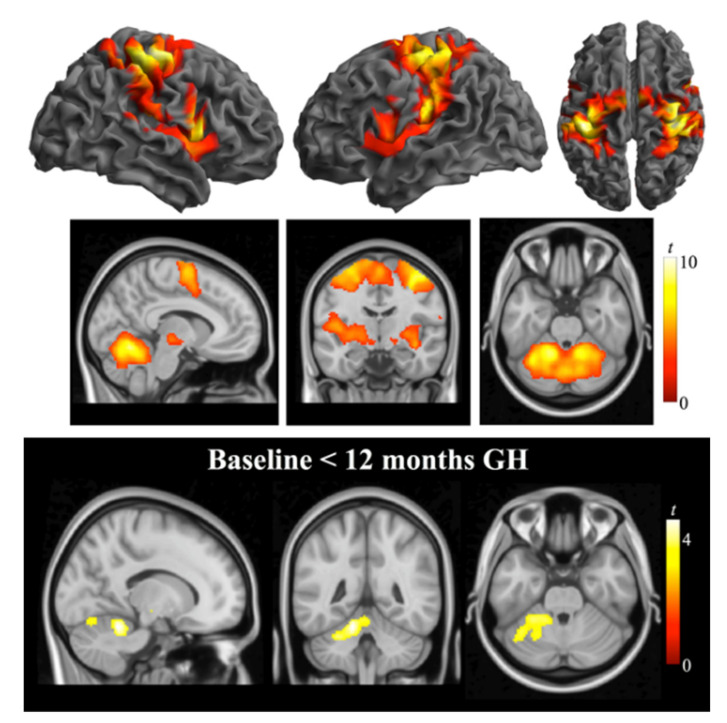
Top and middle rows: Brain activation (one-sample *t*-test) during the bimanual anti-phase flexion-extension task in patients with Prader–Willi syndrome 12 months after the initiation of rhGH treatment. The functional data are displayed on the lateral and dorsal cortical surfaces (white background) and superimposed on a high-resolution anatomical template (sagittal, coronal, and axial views, respectively; black background) using SPM. Bottom row: Differences in whole-brain activation between baseline and 12 months after the initiation of rhGH treatment (paired-sample *t*-test). Activations are thresholded at *P*_FWE_-corrected < 0.05. Color bars represent *t*-values. Right in axial and coronal views corresponds to the right hemisphere.

### 3.3. Correlations

The percentage of change in IGF-I did not correlate with the percentage of change in total body fat, lean body mass, or adjusted appendicular lean mass indices. The increase in IGF-I did not correlate with improvements in the TUG and BBS tests, and these two tests did not correlate with the increase in lean mass.

Voxel-wise regression analyses of imaging data revealed no significant correlations between changes in brain activation and improvement in the TUG or BBS tests or percentage of change in body composition parameters. The percentage of change in IGF-I was not significantly associated with the change in brain activation in the cerebellum; in the whole-brain analysis, the only correlation found with a change in brain activation was a negative correlation in a small cluster located in the brainstem (peak at x = 0, y = −34, z = −14; *t* = 5.77) during the bimanual coordination task.

## 4. Discussion

In the present study, we found that one year of rhGH treatment in adult patients with PWS improved body composition, functional mobility, and balance, and increased functional activation in the cerebellum.

### 4.1. Body Composition

Our findings are in line with those of other studies, where patients with PWS usually have a lower percentage of lean body mass and a higher ratio of fat to lean mass than both normal-weight subjects and obese subjections without PWS [9], representing a unique model of sarcopenia [38]. In our study, 12 months’ treatment with rhGH improved body composition but did not reverse the fat-to-lean mass ratio; these findings corroborate those of other studies with similar [23] or longer [18] duration of rhGH treatment. Interesting, in a recent cross-sectional study in 22 adult patients with PWS, Sjöström et al. [39] showed that long-term rhGH treatment (median, 20 years) was able to reverse the fat-to-lean mass ratio. All 22 patients in their study had greater fat-free mass than fat mass, although the difference between fat-free mass and fat mass was less pronounced in women than in men. They observed no important side-effects, so their findings support the long-term use of rhGH treatment. In our study, we observed no reduction in weight or BMI after one year of rhGH treatment. These results agree with those reported in most studies, although a recent systematic review [40] reported that some studies have found a significant decrease in BMI after two years of rhGH treatment in adults with PWS [22]. Taken all together, these results suggest that these anthropometric measures are neither sufficient nor representative of the improvement in body composition that occurs with rhGH.

### 4.2. Motor Performance

Our findings at baseline corroborate those of other studies, where subjects with PWS have less muscle strength than both healthy subjects and obese subjects without PWS [9]. The repercussions of this deficit are reflected in the high negative correlation between patients’ handgrip strength and caregivers’ bodily pain [7]. Contrary to our expectations, we found no increase in grip strength after rhGH therapy. Our results differ from those reported by Lafortuna et al. [18], who found grip strength increased 7% after 12 months’ rhGH treatment and 13% after 24 months’ rhGH treatment in a group of 15 obese individuals with PWS, independently of GH secretory status (40% had severe GHD). Other authors assessing muscle strength with other methods have reported varied results [21,41].

In addition to muscle mass, various factors may have influenced our patients’ results in the handgrip test. Grip strength correlates substantially with hand length in patients with PWS [42]. In fact, one of the subjects in our cohort could not perform the task because her hand was too small to grip the dynamometer. Additionally, subjects may have received insufficient pre-test training, although Hsu et al. [43] reported that, on its own, a three-month training program did not significantly improve grip strength in a small sample of subjects with PWS [43]. Finally, young adults with PWS have cognitive and behavioral problems [44] that may adversely affect compliance with instructions and attention while doing the test.

Adults without balance problems can complete the TUG test in <10 s, but those with limited mobility skills take > 30 s [45]. A TUG test time < 20 s indicates that the individual is independent for basic transfers [35]. In our study, at baseline, none of the subjects with PWS took > 15 s to complete the TUG test, although the PWS group required more time to complete the test (5.16 s vs. 8.76 s in the age-matched healthy sample, *p* < 0.001); these results are in line with those reported by Chiu et al. [7], who did not take into account GH secretory status or rhGH treatment. On the other hand, higher BMI has been associated with increased functional deterioration, which could lead to impaired balance and an increased risk of falls [28]. Moreover, in addition to excessive body weight, muscle weakness may impair balance in subjects with PWS [46]. Using static posturography, which measures the center of pressure, Capodaglio et al. [46] found worse balance ability in subjects with PWS than in obese subjects without PWS or healthy individuals.

Twelve months after starting rhGH treatment, the time our cohort of patients required to complete the TUG test was significantly lower, suggesting improvements in agility and balance. These results contrast with those of other studies. A randomized controlled study in Scandinavia assessed balance before and after two years of rhGH treatment, finding no differences between the two measurements, despite improvements in body composition [22]. However, rather than the TUG test, that study used the Guralnik test extended to 10 m [47]. Similarly, Lafortuna et al. [18] found no improvement in agility with the modified 10 m walking test after two years of rhGH treatment.

The TUG test has also been correlated with the BBS [48], a scale that assesses the ability to maintain balance through a sequence of exercises of varying difficulty. As in the TUG test, the PWS group scored significantly lower than the control group in the baseline assessment. Nonetheless, only one patient with PWS scored below the cutoff commonly used to identify an increased risk of falls [49]. Our cohort’s BBS score also improved significantly after one year of rhGH treatment. Although we found no correlation between changes in IGF-I and changes in BBS scores, rhGH treatment seems to have directly or indirectly influenced our cohort’s performance, although we cannot rule out the possible influence of familiarity with the tasks in the assessment.

### 4.3. fMRI

To our knowledge, this is the first study to explore brain activity with fMRI before and after rhGH therapy in patients with PWS. In an earlier study, our group reported weaker cerebellar activation in adults with PWS and GHD compared with controls during the performance of the motor tasks before rhGH therapy, especially in tests involving coordination of both hands and fine finger movements [16]. These findings are consistent with the characteristic difficulties people with PWS have in manual dexterity and finger coordination [25,43] as well as with the smaller cerebellar volume and relative amount of white matter in this region reported in children and adults with PWS [17,50,51,52,53].

Twelve months after the initiation of rhGH treatment, we observed a significant increase in neural activation in the cerebellar cortex involving lobes IV/V and VI during the performance of the bimanual coordination task, tying in with the importance of the anterior cerebellar homunculus in coordinating hand movements [54]. We can only speculate about possible mechanisms by which rhGH therapy might increase motor-related neural activation. The cerebellum has GH receptors, and GH is necessary for cerebellar development [55]. Interestingly, children with GHD have functional alterations in cerebellar networks whereas children with idiopathic short stature do not [56]. Furthermore, patients with other conditions, such as Down’s syndrome, who were treated with GH showed better fine-tuning control of movement, and this difference was attributed to the action of GH and IGF-I in the cerebellum [57]. Nevertheless, in our study we found no correlation between the increase in IGF-I and neural activation in the cerebellum or with improvements in TUG and BBS scores, possibly due to the small size of our sample.

### 4.4. Myokines

To gain insight into muscle function in individuals with PWS before and after rhGH treatment, we explored plasmatic levels of the myokines irisin and myostatin. Baseline levels of irisin in plasma were significantly higher in the PWS group than in the control group. Hirsch et al. [32] found salivary irisin was markedly elevated in adults with PWS compared with healthy controls matched by BMI, but no differences in irisin levels after resistance exercise in a study of 11 young patients with PWS [58]. Our findings are at odds with those reported by Faienza et al. [31], who found no difference in irisin levels between controls and a sample of 73 subjects with PWS, 52 of whom were adults, and with those reported by Mai et al. [30], who found similar levels in obese individuals with PWS and lean healthy controls, but higher levels in obese individuals without PWS [30].

Previous work has suggested that muscle mass is the main determinant of circulating irisin levels in humans [59], but these levels can be influenced by genetic variations among obese individuals [30]. Indeed, it has been hypothesized that pathological conditions like obesity render adipose tissue and metabolic dysfunction even more relevant for irisin production than other body tissues [60]. Thus, the relative contributions of muscle and adipose tissue to irisin production might differ depending on the pathophysiological setting, adipose tissue dysfunction, and/or the fat-to-lean mass ratio, which could explain why, despite having less lean mass than the controls, our subjects with PWS do not have lower irisin levels than the controls.

Myostatin plays an inhibitory role in muscle development [27]. Several lines of evidence demonstrate that obesity is associated with increased myostatin expression [61]. We found similar baseline levels of myostatin levels in the two groups, and these levels did not change significantly after 12 months’ rhGH treatment. These results are in line with those reported by Castro-Gago et al. [29], who found no differences in basal myostatin levels in five patients with muscle disease, one of whom had PWS, with respect to the control group.

One study found that IL-6 is the first myokine secreted into the bloodstream in response to muscle contractions [62]. In our study, we found no baseline differences in IL-6 levels, and the levels of this cytokine remained unchanged after rhGH therapy. Studying the levels of several myokines, including IL-6, after endurance exercise in 11 individuals with PWS, Hirsch et al. [58] found no significant differences between cases’ and controls’ baseline levels and a significant increase after exercise in both groups. In contrast, an earlier small study by our group found higher levels of plasma markers of low-grade systemic inflammation, including IL-6, in patients with PWS than in obese and lean controls, and this difference persisted postprandially [63].

Among the major strengths of the present study is the comparison of baseline measurements in our cohort of PWS patients against a control group of healthy volunteers matched for age and sex. Remarkably, there were no dropouts during the study period. Finally, to our knowledge, this is the first study to incorporate fMRI to explore brain activation of motor function in adults with PWS before and after rhGH therapy.

Among the limitations of our study is the lack of an age-matched control group of obese individuals. Thus, it is impossible for us to gauge the extent to which some observations might depend on the state of obesity rather than on PWS per se. Moreover, we did not randomize individuals in the PWS group to receive a placebo or rhGH treatment. Moreover, our sample was small, although the low prevalence of PWS makes it difficult to conduct large-scale studies. Another aspect that warrants caution is that we used DXA to assess body composition; DXA is a validated technique, but it has limitations in patients with extreme obesity. Finally, none of the tests used to assess motor function was designed for patients with PWS, and this may have influenced their execution and results.

## 5. Conclusions

Our study found that one year of rhGH treatment in adult patients with Prader–Willi syndrome was associated with improved body composition, functional mobility, and balance, but did not increase handgrip strength. This treatment had only minimal, solvable effects on blood glucose and on sleep apnea–hypopnea. Moreover, fMRI showed favorable changes in cerebellar activation during motor tasks. Thus, our results add to the growing body of evidence that rhGH treatment can help improve functionality in adult patients with PWS. Larger studies are required to corroborate our findings.

## Figures and Tables

**Table 1 jcm-11-01831-t001:** Anthropometric, biochemical, and body composition parameters at baseline in the PWS and control groups.

	Control Group (*n* = 22) Median (IQR)	PWS Group (*n* = 27) Median (IQR)	*p*-Value *
Weight (kg)	65.9 (58.8–71.3)	89.6 (70.5–105.5)	<0.0001
BMI (kg/m^2^)	22.3 (21.2–22.6)	34.5 (31.1–41.3)	<0.0001
WC (cm)	80 (72–83)	110 (101–124)	<0.0001
IGF-I (ng/mL)	226 (193–304)	143 (95–188)	<0.0001
Glucose (mmol/L)	4.36 (4.16–4.88)	5.0 (4.55–6.99)	0.0039
HbA1c (%)	5.1 (4.9–5.3)	5.6 (5.3–6.9)	0.0002
Irisin (ng/mL)	89.8 (41.8–219.4)	982.3 (519.4–1789.6)	<0.0001
Myostatin (ng/mL)	5.44 (3.05–8.1)	4.8 (2.95–7.72)	0.6714
IL-6 (pg/mL)	11.5 (1.69–44.7)	15.4(3.58–66.4)	0.6191
Fat mass (%)	30.8 (28.4–39.0)	56.3 (49.3–61.1)	<0.0001
Lean mass (%)	69.2 (61.0–71.6)	43.7 (38.9–50.1)	<0.0001
ALM/weight	0.30 (0.26–0.32)	0.19 (0.16–0.21)	<0.0001
ALM/BMI	0.80 (0.70–1.02)	0.48 (0.38–0.55)	<0.0001
Handgrip strength (kg)	31 (28–39)	18 (13–22)	<0.0001
BBS (points)	56 (55–56)	53 (49–54)	<0.0001
TUG (seconds)	5.16 (4.74–5.45)	8.76 (7.56–11.9)	<0.0001

PWS: Prader–Willi syndrome; IQR: interquartile range; BMI: body mass index; WC: waist circumference; IGF-I: insulin-like growth factor-1; HbA1c: glycosylated hemoglobin; ALM: appendicular lean mass; BBS: berg balance scale; TUG: timed up and go test. * Student’s *t*-test for normally distributed variables, Mann–Whitney test for non-normally distributed variables.

## Data Availability

Data supporting reported results can be accessed in Parc Taulí server (contact jcoliva@tauli.cat or the corresponding author acaixas@tauli.cat).

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
