# Peer review of "One Year of Recombinant Human Growth Hormone Treatment in Adults with Prader–Willi Syndrome Improves Body Composition, Motor Skills and Brain Functional Activity in the Cerebellum"

_jcm, 2022, doi:10.3390/jcm11071831_

Round 1
Reviewer 1 Report
Thank you for the opportunity to review the manuscript “One year of growth hormone (GH) treatment in adults with Prader-Willi Syndrome improves body composition, motor skills and brain functional activity in the cerebellum” by Laia Casamitjana et al. It is an interesting and valuable study on adult patients with PWS treated with recombinant human growth hormone (rhGH).
However, there are some issues that need reconsideration.
Please find my recommendations for improvement the manuscript below.
- Please use a term recombinant human growth hormone (rhGH). In the title the abbreviation for rhGH is not necessary.
- Please use a term “patients/adults/subjects with PWS” instead of “PWS patients/adults/subjects” throughout the paper. Similarly, line 391”children with GHD”, not “ GHD children”.
- There is lack of references for GHD stimulation tests interpretation.
- Do I understand correctly that 2 patients had a nonspecific molecular diagnosis? This needs to be explain.
- What were the maximal rhGH doses for females/males?
- The control group- please add a total n.
- AHI- there is a short mention in the Table 2 and in the Results and I believe it should be explain in much more detail, and added as well in the Methods section (what kind of polysomnography, details of the results- obstructive vs central apneas etc.).
- Results- some of them are unnecessary repeated in the text and in the tables.
- Patients with DM- please add more details (worsening of HbA1C? what kind of changes of the treatment was implemented?).
- Discussion, lines 305-306: “A recent cross-sectional study showed a normal relationship between FM and LM assessed by bioimpedanciometry 20 years after GH treatment in a cohort of 22 adult PWS subjects [36].- please cite the ref. in more details.
- Lack of mentioning the strengths of the research.
Author Response
Dear reviewer,
Please find uploaded the changes required. We think that the manuscript has now improved.
Thank you for the opportunity to review the manuscript “One year of growth hormone (GH) treatment in adults with Prader-Willi Syndrome improves body composition, motor skills and brain functional activity in the cerebellum” by Laia Casamitjana et al. It is an interesting and valuable study on adult patients with PWS treated with recombinant human growth hormone (rhGH).
However, there are some issues that need reconsideration.
Please find my recommendations for improvement the manuscript below.
Dear reviewer I,
Thank you for highlighting the value of this work. The contributions you have made will certainly help us to improve the quality of the manuscript. Here below, we respond to each comment.
- Please use a term recombinant human growth hormone (rhGH). In the title the abbreviation for rhGH is not necessary: Thank you for the suggestion. We have changed this abbreviation throughout the text.
- Please use a term “patients/adults/subjects with PWS” instead of “PWS patients/adults/subjects” throughout the paper. Similarly, line 391”children with GHD”, not “GHD children”. Thank you for pointing this out. This has also been changed throughout the manuscript.
- There is lack of references for GHD stimulation tests interpretation. The reference has been added, line 105.
- Do I understand correctly that 2 patients had a nonspecific molecular diagnosis? This needs to be explain. Indeed, one patient had an atypical deletion from locus MICRN3 (BP2) to APAB2 (BP4). The other missing item was a subject with a type I deletion. Both items have been corrected in the manuscript, line 234.
- What were the maximal rhGH doses for females/males? The maximal dose for males was 0.6 mg/day and for women 0.8 mg/day. These data have been added to the manuscript, line 120
- The control group- please add a total n. This has been clarified in line 231
- AHI- there is a short mention in the Table 2 and in the Results and I believe it should be explain in much more detail, and added as well in the Methods section (what kind of polysomnography, details of the results- obstructive vs central apneas etc.).
We have added the required information in the Methods section and more details in the results section.
- Results- some of them are unnecessary repeated in the text and in the tables. Thank you for pointing this out. Repeated data have been removed.
- Patients with DM- please add more details (worsening of HbA1C? what kind of changes of the treatment was implemented?).
In order to clarify this question, we have added the following paragraphs in lines 235-237 and 267-269.
“Six patients had type 2 diabetes with a good glycemic control (HbA1c<7.5%); of these, one was treated only with metformin, two with metformin plus DPPIV inhibitor plus insulin secretagogue, and three with metformin plus DPPIV inhibitor plus insulin”
“All six patients with T2DM required treatment adjustments because metabolic control slightly worsened. We had to add a second oral agent in one, start insulin treatment in two of them and increase insulin dose 0.1 U/Kg/day in three of them. These adjustments could keep straight control”
- Discussion, lines 305-306: “A recent cross-sectional study showed a normal relationship between FM and LM assessed by bioimpedanciometry 20 years after GH treatment in a cohort of 22 adult PWS subjects [36].- please cite the ref. in more details.
We have given more details about results from this reference in the discussion
- Lack of mentioning the strengths of the research.
We have added a paragraph on this point in lines 456-460:
“Among the major strengths of the present study is the comparison of baseline measurements in our cohort of PWS patients against a control group of healthy volunteers matched for age and sex. Remarkably, there were no dropouts during the study period. Finally, to our knowledge, this is the first study to incorporate fMRI to explore brain activation of motor function in adults with PWS before and after rhGH therapy”
Reviewer 2 Report
This manuscript investigates a beneficial effect of one-year growth hormone treatment in adults with Prader-Willi syndrome in terms of not only stature (being short) stature but also body composition, muscle strength, and motor function by doing a multiparametric analysis based on a wide variety of measurements ranging from blood chemistry to functional magnetic resonance imaging.
The hypothesis is relevant since such a contribution to motor function and coordination of GH needs to be elucidated upon for better clinical intervention. However, I have to address the fact that there are many serious concerns including a lot of minor points.
Major concerns
General impression
Frankly speaking, I have to comment that the quality of the current version of the manuscript seems akin to a first draft or summary, for the following reasons: 1) there is a lack of an appropriate amount of full sentences in paragraph form; 2) there are bullet points and merely listed results; 3) there are mixed discussions without the findings being prioritized, and 4) there is an inconsistency between the hypothesis and the conclusion.
Introduction
The paragraphs between the 1st and last ones need to be written coherently for interested readers to better understand the reason why the authors selected such a series of examinations.
Methods
The demographics of participants should be moved to the results, although inclusion or exclusion criteria and ethical considerations need to be placed here.
Results
The tables need to have added the numbers of participants in each group.
Discussion
It is quite difficult for interested readers to understand what the authors would like to show since each discussion is just listed in sequence and is redundant. I feel that the lack of integrated interpretation of what each result means and how they are related significantly reduce the value of this manuscript.
Minor points
There are too many typos, misuses of abbreviations, and misuse of large or small characters to comment on.
In summary, the authors should reconsider these points to make the study relevant and show how it can be informative. I would appreciate it if the authors could recheck these minor points not to discard the priority of this valuable manuscript.
Author Response
This manuscript investigates a beneficial effect of one-year growth hormone treatment in adults with Prader-Willi syndrome in terms of not only stature (being short) stature but also body composition, muscle strength, and motor function by doing a multiparametric analysis based on a wide variety of measurements ranging from blood chemistry to functional magnetic resonance imaging.
The hypothesis is relevant since such a contribution to motor function and coordination of GH needs to be elucidated upon for better clinical intervention. However, I have to address the fact that there are many serious concerns including a lot of minor points.
Dear Reviewer II,
Thank you for highlighting the potential value of this work. We have worked on the listed concerns to try to make this manuscript clearer and improve its quality.
Major concerns
General impression
Frankly speaking, I have to comment that the quality of the current version of the manuscript seems akin to a first draft or summary, for the following reasons: 1) there is a lack of an appropriate amount of full sentences in paragraph form; 2) there are bullet points and merely listed results; 3) there are mixed discussions without the findings being prioritized, and 4) there is an inconsistency between the hypothesis and the conclusion.
In this new version we have worked on these points
1)There is a lack of an appropriate amount of full sentences in paragraph form
We have improved the writing, now with full sentences and structured paragraphs, throughout the manuscript (mainly in the results section)
2) There are bullet points and merely listed results
We eliminated bullets throughout results section (sections 3.1 and 3.2)
3) There are mixed discussions without the findings being prioritized
We have improved the discussion by joining some paragraphs and grouping them into sections, so as to make them easier to be read. We have also tried to highlight the most relevant results ( pages 8 to 11)
4) There is an inconsistency between the hypothesis and the conclusion.
We tried to ensure consistency between the hypotheses and conclusions.
- Hypothesis ( introduction section, page 2) :“We sought, first, to characterize body composition, strength, balance, myokines, and brain neural activation during the performance of different motor tasks in adults with PWS and GHD by comparing these parameters with healthy controls and, second, to determine the effects of one year of rhGH treatment on these parameters by comparing baseline and posttreatment values. We hypothesized that treatment would improve body composition (increasing lean mass and decreasing fat mass), increase balance and strength, and improve brain neural activation during motor tasks”
- Conclusions ( Conclusions section, page 11): “Our study found that one year of rhGH treatment in adult patients with Prader-Willi syndrome was associated with improved body composition, functional mobility, and balance, but did not increase hand-grip strength. This treatment had only minimal, solvable effects on blood glucose and on sleep apnea-hypopnea. Moreover, fMRI showed favorable changes in cerebellar activation during motor tasks. Thus, our results add to the growing body of evidence that rhGH treatment can help improve functionality in adult patients with PWS. Larger studies are required to corroborate our findings.
Introduction
The paragraphs between the 1st and last ones need to be written coherently for interested readers to better understand the reason why the authors selected such a series of examinations.
In this second version we have tried to better explain the points on which our study is based in order to make them easier to understand. We have rewritten the introduction section.
Methods
The demographics of participants should be moved to the results, although inclusion or exclusion criteria and ethical considerations need to be placed here.
The demographics of participants have been removed from the methods section and replaced in the Results one as suggested, in line 231.
Results
The tables need to have added the numbers of participants in each group.
The number of patients in each group has been included in table 1.
Discussion
It is quite difficult for interested readers to understand what the authors would like to show since each discussion is just listed in sequence and is redundant. I feel that the lack of integrated interpretation of what each result means and how they are related significantly reduce the value of this manuscript.
We have rewritten the discussion. We also have joined some paragraphs and grouped them into sections, so as to make them easier to be read. (pages 8 to 11)
Minor points
There are too many typos, misuses of abbreviations, and misuse of large or small characters to comment on.
Sorry about that. The revised manuscript presented here has been corrected by an experienced medical editor from the USA.
In summary, the authors should reconsider these points to make the study relevant and show how it can be informative. I would appreciate it if the authors could recheck these minor points not to discard the priority of this valuable manuscript.
We have reconsidered all the required points and we thank the reviewer for the possibility of improving the manuscript.
Round 2
Reviewer 2 Report
Thank you for your corrections on your manuscript.
I think that the quality of the revised version has reached to the acceptable level.
Finally, please recheck the size of fonts in the tables, as it seems to be different between the tables.
Author Response
Dear Reviewer,
Thank you for considering that the revised version has reached to the acceptable level.
The minors changes you required in this second revision were:
1) Finally, please recheck the size of fonts in the tables, as it seems to be different between the tables.
We have changed the size of fonts in table 1, to the same in tables 2 and 3.
We really appreciate your revision.